# Genesis and propagation of exogenous sediment pulses in mountain channels: insights from flume experiments with seismic monitoring

Marco Piantini[1,2], Florent Gimbert[1], Hervé Bellot[2], Alain Recking[2]

[1] University Grenoble Alpes, CNRS, IRD, Institute for Geosciences and Environmental Research (IGE), Grenoble, France

[2] University Grenoble Alpes, INRAE, ETNA, Grenoble, France

*Correspondence to*: Marco Piantini (marco.piantini@univ-grenoble-alpes.fr)

**Abstract.** In the upper part of mountain river catchments, large amounts of loose debris produced by mass wasting processes can accumulate at the base of slopes and cliffs. Sudden destabilizations of these deposits are thought to trigger energetic sediment pulses that may travel in downstream rivers with little exchange with the local bed. The dynamics of these exogenous sediment pulses remains poorly known because direct field observations are lacking, and the processes that control their formation and propagation have rarely been explored. Here we carry out flume experiments with the aims of investigating (i) the role of sediment accumulation zones in the generation of sediment pulses, (ii) their propagation dynamics in low-order mountain channels, and (iii) the capability of seismic methods to unravel their physical properties. We use an original setup where we supply with liquid and solid discharge a low slope storage zone acting like a natural sediment accumulation zone, and connected to a downstream 18 % steep channel equipped with geophones. We show that the ability of the self-formed deposit to generate sediment pulses is controlled by the sand content of the mixture. In particular, when a high fraction of sand is present, the storage area experiences alternating phases of aggradation and erosion strongly impacted by grain sorting. The upstream processes also influence the composition of the sediment pulses, which are formed by a front made of the coarsest fraction of the sediment mixture, a body composed of a high concentration of sand corresponding to the peak of solid discharge, and a diluted tail that exhibits a wide grain size distribution. Seismic measurements reveal that the front dominates the overall seismic noise, but we observe a complex dependency between seismic power and sediment pulses' transport characteristics, which questions the applicability of existing seismic theories in such context. These findings challenge the classical approach for which the sediment budget of mountain catchments is merely reduced to an available volume, since not only hydrological but also granular conditions should be considered to predict the occurrence and propagation of such sediment pulses.

## 1 Introduction

Sediment transport processes play a key role in fluvial geomorphology (Schumm, 2003) and natural risk management (Badoux et al., 2014), since they exert a major control in the intensity with which rivers can impact the landscape and the safety of inhabited regions. This is particularly evident in mountain catchments, where catastrophic floods are exacerbated by a rapid hydrological response to rainfall (high hydrological connectivity, (Wohl, 2010)) and a large mobilization of sediments (Recking, 2014). Predicting when and how sediments move throughout mountain channels, however, remains challenging since onset of motion criteria and bedload transport laws have mostly been established for lowland rivers and have limited applicability to mountain environments (Schneider et al., 2016). Mountain rivers are characterized by a wide range of morphological units whose peculiarities cannot be neglected when studying sediment transport (Lee and Ferguson, 2002; Comiti et al., 2009; Zimmermann et al., 2010). For instance, several works have shown that large-scale bed roughness are expected to affect bed shear stress (Bathurst et al., 1983; Wiberg and Smith, 1991; Solari and Parker, 2000; Lamb et al., 2008; Recking, 2009; Prancevic and Lamb, 2015), and grain sorting processes have a stronger impact in producing bedload fluctuations compared to lowland streams (Recking et al., 2009; Bacchi et al., 2014). Moreover, the steepness of mountain channels may help trigger debris flows, which are energetic transport processes where the sediment concentration is so high (greater than 50 % by volume) that the solid phase influences the behaviour of the flow as much as the fluid phase (Iverson, 1997). The conditions of transition from bedload to debris flow remains debated partly due to lacking field observations (Mao et al., 2009; Prancevic et al., 2014).

For both fluvial and debris flows processes, in addition to the hydrological forcing, sediment supply conditions play an important role (Benda and Dunne, 1997; Bovis and Jakob, 1999; Recking, 2012) and their spatial and temporal variabilities add complexity to predictions. Mountain channels that are coupled to sediment production zones (high landscape connectivity, (Wohl, 2010)) are particularly prone to receive episodic inputs of material coming from upstream sections of the catchment, where sediments produced by mass wasting processes accumulate in the form of talus slope or along low-slope stretches as loose scree deposits. However, this storage is often temporary, since rainfall and runoff descending from upper slopes can destabilize these accumulation zones and trigger sediment transport towards downstream channels (Berti et al., 1999; Fontana and Marchi, 2003; Gregoretti and Fontana, 2008). This is for example the case in the Roize River, France (Fig. 1a). The upper part of the catchment is characterized by cliffs producing a large amount of debris that accumulate at the slope's toe (Fig. 1b and Lamand et al., 2017), and as a result of hydrological and gravitational phenomena, sediments are occasionally released to the coupled reach (Fig. 1c) where they are transported downstream to a reception zone (sediment trap). The dynamics of transport throughout the river reach has been shown to be strongly related to the activity of these headwater sediments sources (Piton and Recking, 2017). Thanks to exogenous inputs of sediments, such streams can suddenly switch from supply limited to overcapacity conditions, as illustrated in Fig. 2 where the non-alluvial and inactive bed of the Ruisseau de la Gorge (French Alps) suddenly experienced a large transport event in 2015. As the transported sediments were much finer than the bed in place, an upstream and exogenous input of the material was suggested.

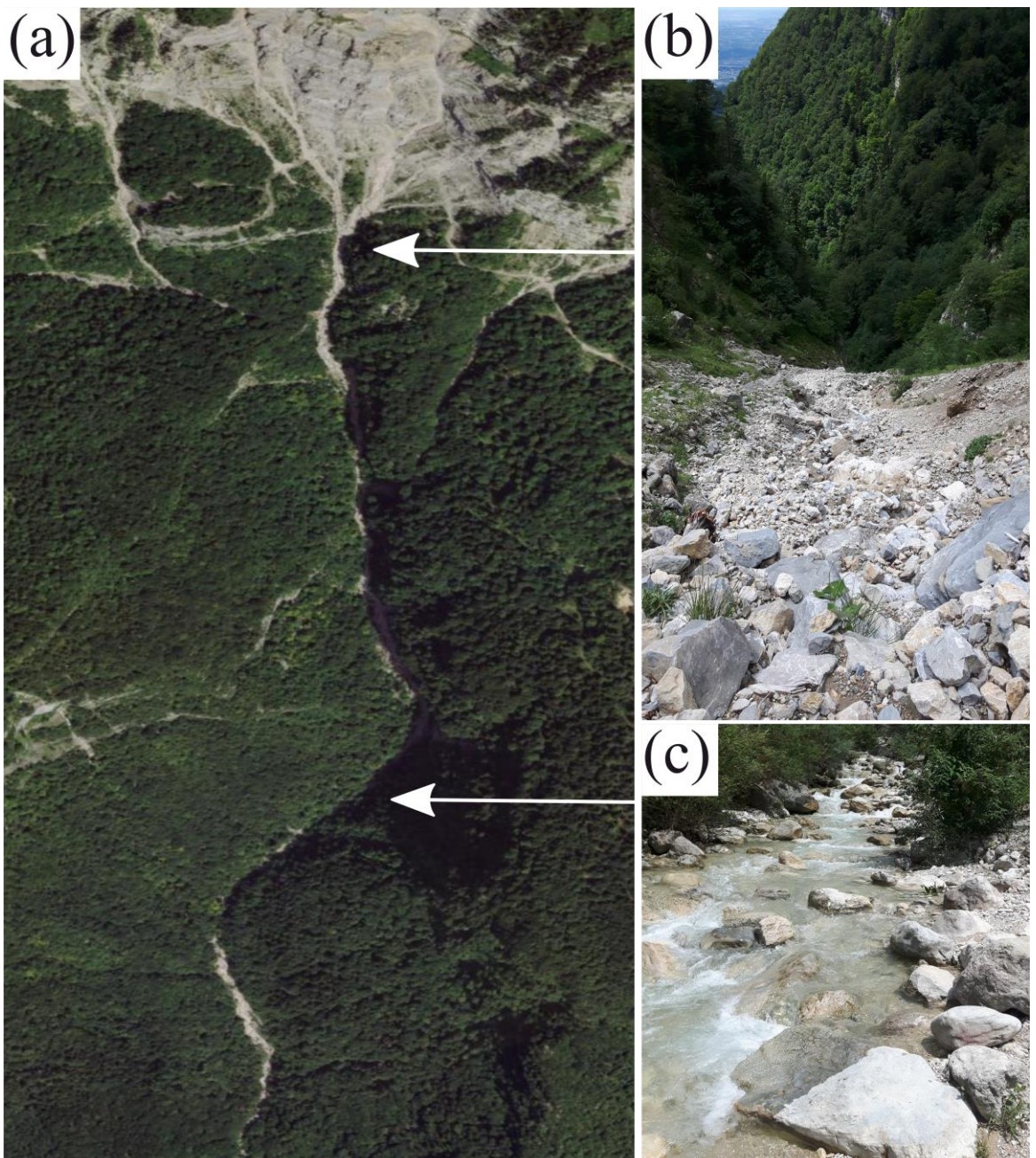

**Figure 1: (a) A typical mountain stream configuration (the Roize River, https://www.geoportail.gouv.fr) with: (b) a production zone (sediments deposits are several meters thick and show evidences of large incisions) and (c) a transfer zone consisting in a narrow steep step-pool morphology.**

Several works have shown that exogenous sediment inputs in a river usually take the form of *sediment pulses*, defined in the

literature as disturbances in bed elevation that propagate downstream translating as a coherent wave and/or dispersing in place

(Sutherland et al., 2002; Brummer and Montgomery, 2006). Previous studies have investigated the evolution of these sediment

pulses in gravel-bed rivers characterized by a maximum slope of 1 %, where the streambed has been shown to actively interact with the injected material (Lisle et al., 1997; Sutherland et al., 2002; Cui et al., 2003; Cui and Parker, 2005; Sklar et al., 2009).

However, low-order mountain rivers usually present geological controls such as rarely mobile boulders and bedrock outcrops, as well as much steeper slopes. In this context, sediment pulses are expected to be transported downstream with a marginal morphological impact on the underlying bed, following the "travelling bedload" concept (Piton and Recking, 2017). To the best of our knowledge, there are no experimental studies that investigate sediment pulses' propagation in such configuration, and the few post-event field observations do not provide information about their spatial and temporal dynamics. Classical

monitoring methods reveal scarce effectiveness for observing pulse-like events (Mao et al., 2009), and therefore sediment pulses are challenging to track due to their localized and potentially energetic nature. In this context, seismic methods represent a robust alternative for providing a non-invasive and continuous monitoring of torrential processes (Burtin et al., 2016) and catastrophic floods (Cook et al., 2018). As sediment transport generates ground vibrations, mechanistic models have been defined to understand the links between river processes and the generated seismic noise (Tsai et al., 2012; Gimbert et al., 2014;

Lai et al., 2018; Farin et al., 2019). Applicability of seismic theories for bedload under relatively low transport rate has been demonstrated in the laboratory (Gimbert et al., 2019) and in the field (Bakker et al., 2020). Seismic models for more concentrated sediment flows have also been tested in the laboratory, in the context of dry granular flows (Arran et al., 2021), and in the field, in the context of debris flows (Zhang et al., 2021). However, the extent to which existing theories apply to a variety of sediment transport flows including sediment pulses, which may lie in between bedload transport and debris flows,

remains to be investigated.

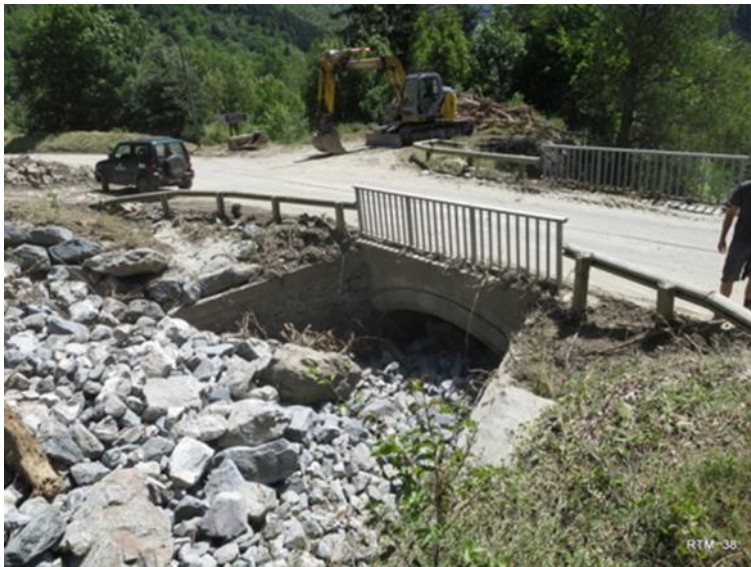

**Figure 2: Effect of a sediment pulse at a bridge section of the Ruisseau de la Gorge (France), a stream that was known by local engineers for having been inactive for decades. The transported material was much finer ($D_{50} = 96\ mm$, $D_{84} = 169\ mm$) than the bed in place ($D_{50} = 250\ mm$, $D_{84} = 413\ mm$).**

In this study we conduct laboratory experiments (i) to explore the role of sediment accumulation zones in the generation of sediment pulses, (ii) to investigate their propagation dynamics in low-order mountain channels, and (iii) to test the capability of seismic methods to infer the flow properties associated to such sediment transport events. We use an original setup where instead of feeding the flume section directly as usually done, we supply with liquid and solid discharge a low slope storage zone connected to the upstream part of a 18 % steep channel. Such an experimental configuration allows us to investigate if a self-formed deposit can generate sediment pulses and how these later propagate in the downstream channel. In $Sect.\,2$ we present the experimental setup and the measurements protocol. Then in $Sect.\,3$ we present our experimental results regarding both the storage area and the channel. Finally, in $Sect.\,4$ we discuss the key results and we describe the main implications for mountain stream morphodynamics.

## 2 Material and methods

### 2.1 Experimental setup and measurements

We use a 6 $m$ long flume made of (i) a 1 m long and on average 0.5 m wide trapezoidal shaped upstream storage area ($\sim 0 - 1\,\%$) and (ii) a 5 m long and 0.1 m wide downstream steep (18 % slope) channel (Fig. 3).

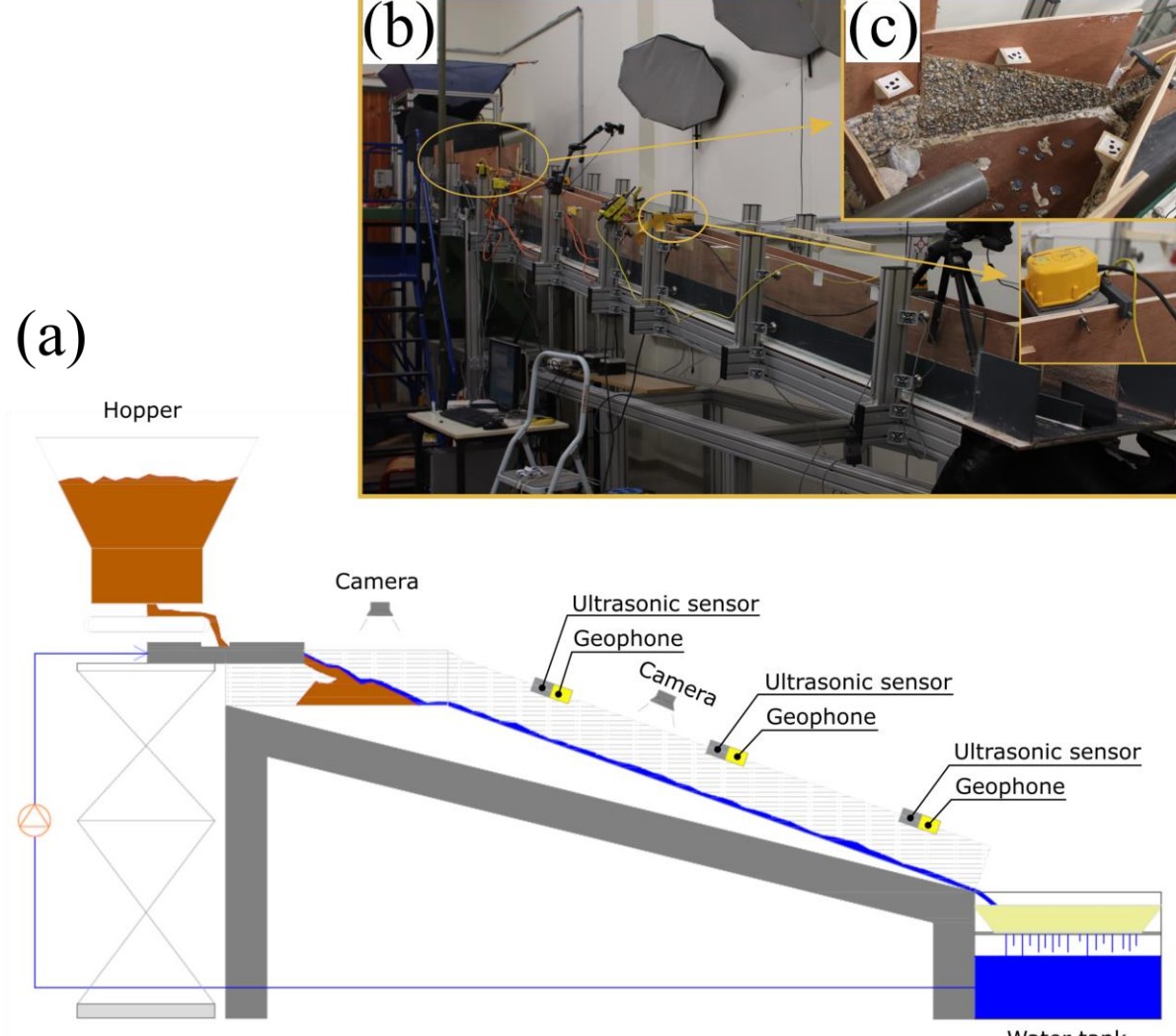

**Figure 3: Panel (a): scheme of the flume with the instrumental equipment; panel (b): a photo of the flume; panel (c): a zoom on (i) the upstream storage area and (ii) one of the three sections equipped with a geophone (yellow device) and an ultrasonic sensor (grey housing).**

Water discharge recirculation is ensured by a pump supplied by a reservoir placed at the flume outlet, whose level is kept constant through an overflow drain. The discharge value is measured with an electromagnetic flowmeter and the flow rate is controlled numerically using a calibrated voltage/discharge relationship. We use a sediment feeding system composed of a hopper connected to a conveyor belt for the solid discharge. The sediment flux is controlled by the velocity of the conveyor belt which is measured by a sensor fixed on one of its rotation axes. As for the water supply we set a calibrated equation in order to regulate the solid discharge from the computer.

The topographic evolution of the storage area is monitored with a sensing camera (Microsoft Kinect) that allows to reproduce a virtual 3-D model from the images through depth-sensing techniques: a light is firstly projected by an infrared sensor, then the reflected pattern is captured to recover the geometry of the object by computing the light's time of flight. The device is used to estimate the volume variation of the deposit and its longitudinal slope.

We video record each experiment with two webcams placed at the inlet section and along the channel (Microsoft HD LifeCam Cinema). Three sections are equipped with a remote transducer ultrasonic sensor (Banner Q45UR Series) having a sampling frequency of 100 Hz and a geophone (3-D Geophone PE-6/B) (Fig. 3) to respectively measure the flow stage and detect flow-induced seismic flume motion generated by particle impacts (Govi et al., 1993). The data from the geophones are recorded on a DATA-CUBE[3] logger with a sampling frequency of 800 Hz. In order to explore the properties of the seismic noise, we compute the power spectral density (PSD) of the signal recorded along the vertical by performing a fast Fourier transform with the Welch's averaging method (Welch, 1967). According to this method the time series is split into overlapping segments (here we chose an overlap of 50 %), and the final PSD results from the average over the PSDs of each segment. We focus on sediment transport-related seismic noise by getting rid of other sources emitted by the experimental device (e.g. water pump, water flow in pipes and on the flume, etc..) through normalizing the raw signal by the seismic power occurring under similar experimental conditions but with no sediment transport (see $Sect. S2$ in Supplement). We measure the sediment flux by sampling the outgoing sediments at the channel exit and we compute the grain size distribution of the samples from sieve measurements. It is worth noting that solid discharge is measured by hand and is consequently not continuous in time, and the sampling frequency is adapted to flow conditions. As flow stage and seismic noise are monitored at a different section than the outlet solid discharge, a time lag between measurements is present. In order to compute the expected temporal delay and to properly compare the measured data, we time shift the outlet solid discharge by estimating the velocity of the flux with a cross correlation between the three flow stage time series. Such a time-shift procedure is appropriate for the seismic analysis thanks to significant signal amplification ($+ 5\ dB$ in average) occurring near the geophone in our experimental setting (see $Sect.\ S4$ in Supplement).

## 2.2 Experimental scaling and input conditions

Although this work does not aim at being the analog of a particular natural prototype, we have built the flume and set the boundary conditions under several scaling considerations. While the dimensionless characteristics of the flume (e.g. slope and sediment transport concentration) can be directly compared to the field, the definition of a scaling parameter is required to estimate the scale reduction of other dimensional parameters of the flume. We follow the approach of Piton (2016) and define a geometrical scaling parameter $\lambda$ as the ratio between a characteristic particle diameter of the natural and experimental river. We choose the $84th$ percentile grain diameter as being considered a proxy of bed roughness, which exerts a major control on river hydraulics:

$$\lambda = \frac{D_{84,natural\ channel}}{D_{84,experimental\ channel}}$$

where $D_{84,natural\ channel}$ is the characteristic particle diameter of the natural channel and $D_{84,experimental\ channel}$ is that of our experimental setup.

Mountain channels are typically characterized by a wide bimodal grain size distribution ranging from fine elements to large boulders provided by an external sediment supply (John Wolcott, 1988; Casagli et al., 2003; Sklar et al., 2017). This is why we choose a bimodal grain size distribution characterized by two modes corresponding to sand ($0.5\ mm < D < 2\ mm$) and cobbles ($4\ mm < D < 8\ mm$) (Table I and Fig. 4) as input. The poorly sorted mixture is obtained with respect to grain size distribution utilized in previous experimental works on steep slope (Bacchi et al., 2014) and is characterized by $D_{50} = 5.16\ mm$ and $D_{84} = 9\ mm$. In order to reproduce the immobile natural roughness of confined bedrock torrents, we glue sediments to the bed and side walls of the flume.

Considering two well-documented steep mountain streams as reference natural channels, the Rio Cordon River (Italy) (Lenzi et al., 2004; Mao and Lenzi, 2007; Schneider et al., 2014) and the Erlenbach River (Switzerland) (Turowski et al., 2009; Schneider et al., 2014), we obtain $\lambda \approx 32$ computed as the average over those two reference streams ($\lambda = 41$ for the Rio Cordon and $\lambda = 23$ for the Erlenbach rivers).

Following the guidelines of Peakall et al. (1996), channel width and length, and depositional height are expected to scale linearly with $\lambda$, while the liquid discharge per unit channel is expected to scale as $\lambda^{1.5}$. Our experimental flume width is thus equivalent to a natural channel width of about 3.2 m, consistent with typical mountain stream widths (see Table 2). The up-scaled channel length corresponds to 160 m, which can be considered as a natural channel reach. The dimensionless experimental slope of 18 % falls within the range of steep mountain streams (see Table 2).

Concerning the upstream storage area, its size is mainly dictated by technical constraints since we use a preexisting steel channel as support for the flume (Fig. 3). Nevertheless, the chosen geometry leads to the formation of a maximum $\approx 0.15$ m thick deposit, which would correspond to a deposit of about 5 m thick in a natural context, consistent with field observations in mountain upper catchments (Berti et al., 1999; Imaizumi et al., 2006). The basal slope in the storage area is arbitrarily set to ~0-1% in order to reduce the transport capacity and let the deposit develop. The influence of storage area's geometry on the observed processes is discussed in $Sect.\ 4.1$.

We chose the flow discharge with respect to standard similitude criteria. In particular we verify that channel's flow conditions are supercritical (Fr>1), fully turbulent (Re>2000) and hydraulically rough (Re*>70) by computing the Froude $Fr$, Reynolds $Re$, and Reynolds particle number $Re^*$, consistently with natural mountain streams (Peakall et al., 1996; Asano and Uchida, 2016). Estimating these parameters requires an estimate of the flow velocity which is computed following Rickenmann and Recking (2011). Finally, considering the above requirements and the flume setup, we prescribe liquid discharge per unit channel width of 0.0045 m²s⁻¹ which is equivalent to about 0.80 m²s⁻¹ in the field. From Schneider et al. (2014), this discharge value is associated with flood events having a recurrence interval of about 5 years for the Rio Cordon and the Erlenbach rivers. The feeding of the upstream storage area set in order to obtain a high solid concentration ($C = 6.7$ %) which corresponds to a hyperconcentrated flow (Coussot and Meunier, 1996).

The main experiment is characterized by an in-channel transport stage $\tau^*/\tau_{CR}^*$ close to 1, where $\tau^*$ is the mean Shields stress

and $\tau_{CR}^*$ the critical Shields stress. We calculate the mean Shields stress as $\tau^* = \frac{\tau}{g(\rho_s-\rho)D_{84}}$, where bed shear stress is

approximated under the assumption of uniform flow as $\tau = \rho u_*^2$, $\rho$ is water density and $u_* = \sqrt{ghS}$ is the bed shear stress

velocity, with $h$ equal to water level, $S$ being the channel slope, $g$ is acceleration due to gravity, $\rho_s$ is sediment density and

$D_{84}$ is the $84th$ percentile particle diameter. The critical shear stress is considered slope dependent and formulated following

Recking et al. (2008) as $\tau_{CR}^* = 0.15 \, S^{0,275}$.

The overall experimental conditions are summarized in Table 1.

**Table 1: Experimental conditions**

| Main Experiment | Reference experiments | Supplementary experiment |
|---|---|---|
| Q $_1$ = 0.45 l s$^{-1}$ | • Varying grain size distribution : | • Without storage area: |
| Q $_s$ = 80 g s$^{-1}$ | 1) Run R1: Uniform fine | 4) Run S1 |
| C = 6.7 % | 2) Run R2: Uniform coarse | |
| Fr = 1.66 | 3) Run R3: Bimodal with a | |
| Re = 2417 | reduced fine fraction | |
| Re$^*$ = 530 | | |
| H / D$_{84}$ = 0.70 | | |
| $\tau^*$ = 0.08 | | |
| $\tau_{cr}^*$ = 0.09 | | |
| $\tau^*/\tau_{cr}^*$ = 0.89 | | |
| Duration = 0.5 h | | |
| Bimodal grain size distribution | | |

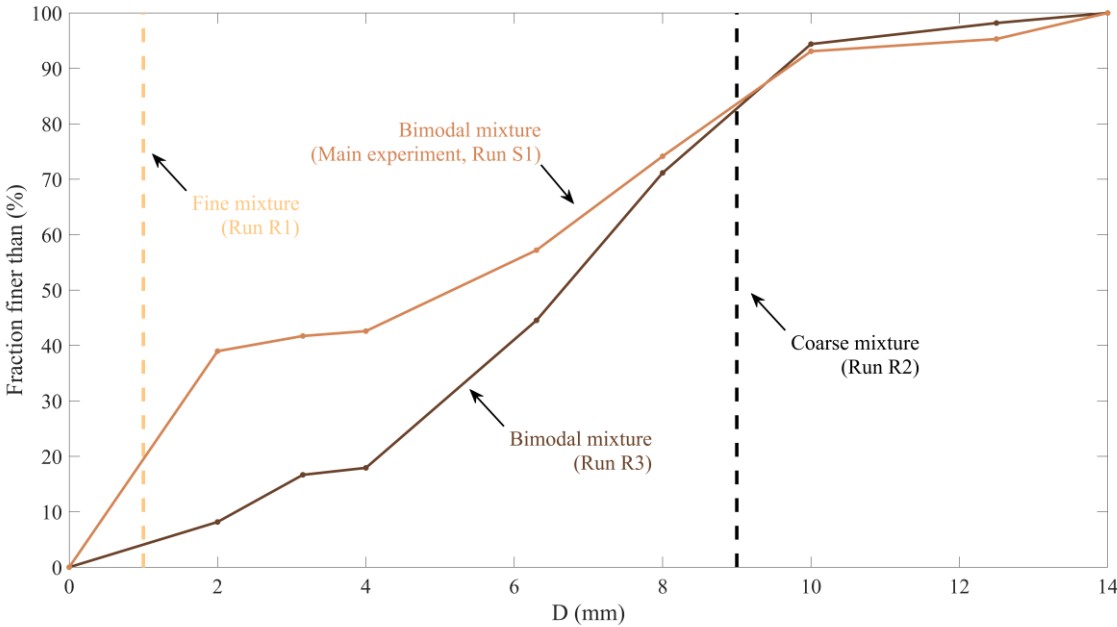

**Figure 4: Grain size distribution of the different sediment mixtures used in the experiments.**

**Table 2: Main characteristics of the Rio Cordon and Erlenbach rivers considered to scale the experimental conditions. The values of liquid discharge for the reference channels refer to a recurrence interval of 5 years (Schneider et al., 2014). The up-scaled**
**experimental values are computed using $\lambda = 32$.**

|  | *Rio Cordon* | *Erlenbach* | *Up-scaled experiments* |
|---|---|---|---|
| $D_{84}$ (mm) | 366 | 206 | 288 |
| Slope (%) | 13.6 | 15 | 18 |
| Width (m) | 5.3 | 3.5 | 3.2 |
| $Q_1$ (m² s⁻¹) | $\approx 1.14$ | $\approx 0.87$ | $\approx 0.80$ |

## 2.3 Additional experiments

In addition to the main experiment, we conduct additional experiments with different grain size distributions in order to explore the effect of grain size heterogeneity on the behaviour of the deposit. We test a bimodal distribution characterized by a reduced amount of sand (20 % less in weight, Run R3 in Table 1 and Fig. 4), and two nearly uniform mixtures characterized by a mean diameter of $1 \, mm$ and $9 \, mm$, respectively Run R1 and Run R2 in Table 1 and Fig. 4. We also carry out a supplementary experiment (Run S1 in Supplement) that consists in feeding the 18 % steep channel directly using the bimodal mixture of the main experiment. Input liquid and solid discharge values are kept constant for each run.

## 3 Results

### 3.1 Dynamics of the deposit in the storage area

The temporal variation of the deposit's volume detected using the Kinect Camera measurements during the Main Experiment is shown with the brown curve on Fig. 5a, while the mechanisms involved in its evolution are investigated through looking at an associated video (Video 1 in the Supplement) and selected images (Fig. 5). During the first minutes (about 5 min), the flow is characterized by a limited transport capacity, which results in nearly total deposition with no sediments reaching the downstream channel. The water flow mainly bypasses the deposit on the sides, although some infiltration also occurs, as attested by subsurface flows coming out of the deposit toe. However, after a while (about 6 min) a large portion of the deposit is submerged, while its upper part experiences a thin but significant surface water flow (Video 1). Local failures efficiently move clusters of sediments at the front of the deposit and on the flanks, such that the deposit grows up in the vertical and horizontal direction until it approaches the connected steep channel. We observe that grains at the surface are preferentially coarse as a result of the downward percolation of finer particles (kinematic sieving, sensu Frey and Church, 2009). These bigger grains create an armour at the surface and also roll to the deposit's toe (yellow bordered particles in Fig. 5c), both processes stabilizing the whole mass. At this stage, the volume reaches its maximum (point 1 in Fig. 5a) with a slope of $\approx 53$ % (brown curve in Fig. 5b) when the armour suddenly breaks and a major *en masse* failure of the deposit is triggered. The armour breaking leads to the formation of a channelized flow that erodes the deposit and transports sediments over a smooth bed of sand, previously hidden in the subsurface (point 2 in Fig. 5a and red bordered area in Fig. 5d). After this first large destabilization that evacuates the eroded material towards the downstream main channel, the deposit reaches its lowest longitudinal slope ($\approx 25$ %) that results in a decreased transport capacity. However, some sediments are still prone to leave the storage area through a small incised channel, such that the total volume does not change significantly (plateau that nearly lasts 300 s after point 2 in Fig. 5a). A new armoured surface starts developing with the formation of bars made of coarse particles, which makes a new aggradation phase possible as the water flow becomes shallow and unchannelized (the sheet flow described by Parker (1998)). The deposit reaches another peak in volume with a heavily armoured surface (point 3 in Fig.

5a and Fig. 5e) before another destabilization occurs. We observe four alternating aggradation and erosion phases until the end of the run, interspersed with minor releases to the channel (see black arrows in Fig. 5a). Aggradation and erosion phases fluctuate between an average deposit's slope of ≈ 48 % (range 45 % - 53 %) and 23 % (range 22 % - 25 %) respectively. The last 1000 $s$ of the experiment are characterized by a generalized depletion of material due to the congestion of the storage area that is no longer able to retain sediments.

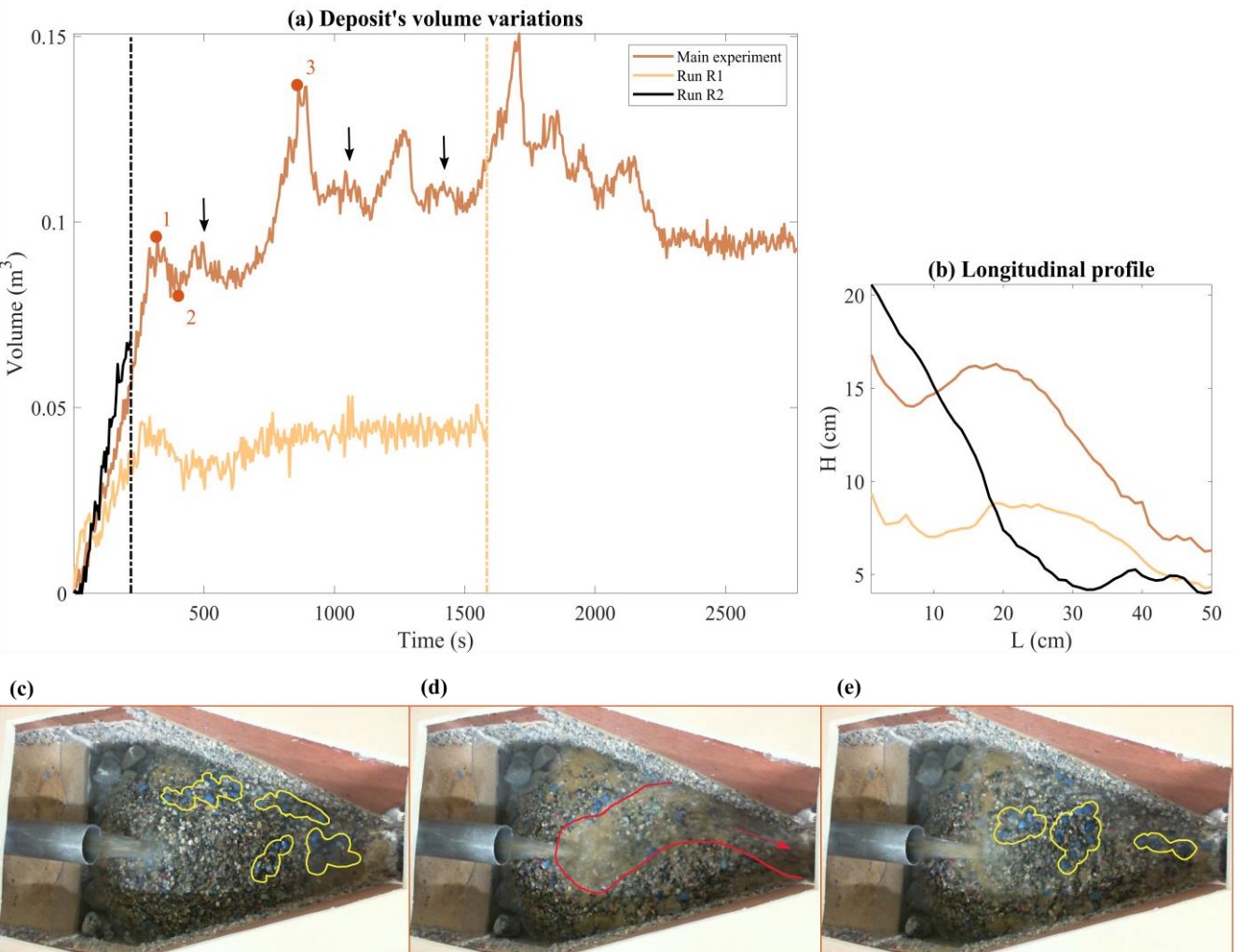

**Figure 5: (a) Results from the Kinect Camera for the three runs. The volume variation of the deposit is shown versus time. The two vertical dotted lines show the end of the runs characterized by a uniform mixture. The black arrows indicate the sediment releases occurring after the larger destabilizations and before the following aggradation phase. The orange dots with the associated number refer to the images presented below: the frames of the video recording represent the steps of the cyclic behaviour experienced by the storage area, with: (c) aggradation phase of the deposit and armouring phase at its maximum; (d) sediment pulse to the channel following the destabilization of the deposit with sand no more hidden but exposed to the flow; (e) new armouring phase. The yellow bordered particles form the surface armour, while the red bordered area shows the destabilized masse. (b) Comparison between the longitudinal profiles of the deposit for the three experiments when the aggradation phase is at its maximum. The profile is the result of the intersection between the deposit and a plane normal to the storage area's base and parallel to the channel.**

Interestingly, we find that the alternating behaviour as described above no longer occurs when using uniform sediment mixtures. The experiment using the mixture of sand (Run R1) first exhibits an aggradation phase during the first 250 s (cream-coloured curve in Fig. 5a) but sand quickly reaches the inlet section of the channel and the storage area starts to release sediments with a mean solid discharge of 156 g s$^{-1}$, before reaching an equilibrium with the inlet solid discharge (see Video 2 in Supplement). The plateau in the cream-coloured curve of Fig. 5 indicates that an equilibrium phase is achieved with no significant deposition or erosion. The experiment carried out with the coarse mixture (Run R2) leads to the formation of a steep pile in front of the injection tube. As the mobility of the grains is low, the deposit grows quickly in the vertical direction and reaches the height of the injection tube long before approaching the channel inlet. Other than for the interlocking effect of the particles, the video recording (see Video 3 in Supplement) clearly shows that the high permeability of the mixture causes the water to fully infiltrate, leading to nearly dry flow conditions at the surface (no water surface flow). We observe a similar behaviour in Run R3 using a bimodal mixture characterized by a low percentage of sand (around 10 % by weight, Fig. 4), whose video recording is presented in Supplement (Video 4) and where the deposit shows a strong stability and no pulses are generated. The different mobility of the three mixtures presented here is materialized by the longitudinal profile computed for each experiment during the maximum extension of the deposit (Fig. 5b). Sand easily reaches the inlet section of the channel and particles are washed away by the flow by preventing the deposit to grow in volume (cream-coloured curve in Fig. 5b). The coarse material is on the other side of the spectrum as the stability of the mixture allows the deposit to reach a 66 % longitudinal gradient (burgundy curve in Fig. 5b). In between these two conditions, the deposit made of the bimodal mixture is able to develop radially thanks to local destabilizations that spread material towards the channel (brown curve in Fig. 5b).

Based on these observations, we hypothesize that, in our experiments, the ability of the deposit to experience alternating phases of storage and erosion with the generation of sediment pulses is controlled by the presence of sand and its downward percolation through the coarser grains. The processes potentially involved are discussed in $Sect. 4.2$.

### 3.2 Sediment pulse's propagation in the downstream channel

We investigate the propagation and physical characteristics of the sediment pulses with a specific experiment focused on the channel having the boundary conditions of the Main Experiment (see Table 1). We use the middle section's ultrasonic and geophone sensors, as well as the hand-made measurements of sediment flux and grain size distribution at the channel outlet. After the time shifting procedure (see $Sect. 2.1$), we find a clear correlation between flow stage and solid discharge measurements (Fig. 6): the passage of solid discharge pulses in the downstream channel is materialized by distinct peaks of about 60 s in the flow stage measurement time series (Fig. 6a). The biggest peaks are associated with a solid discharge of about 340 g s$^{-1}$ (Fig. 6b), which is up to four times larger than the prescribed solid input of 80 g s$^{-1}$, and a sediment concentration that reaches 26.8 % in volume. The magnitude of the sediment pulses is controlled by the dynamics of the upstream storage area, as confirmed by the supplementary experiment (Run S1 in Supplement and Video 5) in which we feed the 18 % steep channel directly with the same bimodal sediment mixture and observe no significant solid discharge

fluctuations. The second solid discharge peak around $t = 700$ s is smaller than the others, since its height is ~1 cm and its mean solid discharge is almost equal to the prescribed solid input ($Q_S = 84$ g s$^{-1}$). We find that this pulse is the result of a sediment release occurring just before the second cycle of aggradation/erosion in the upstream storage area (see $Sect.$ 3.1).

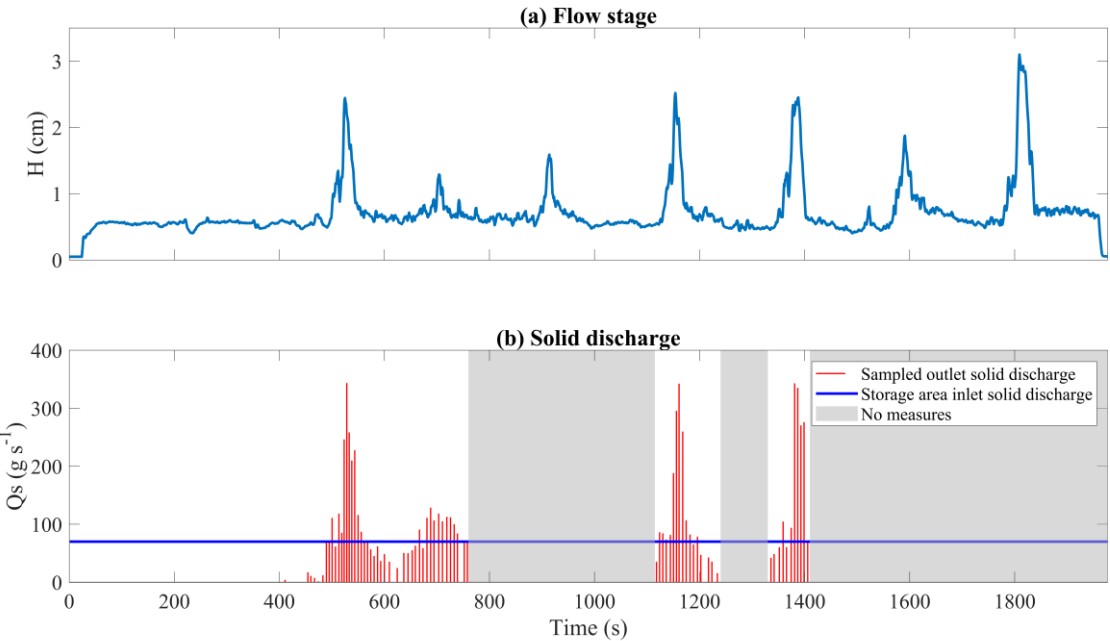

**Figure 6: In-channel measurement time series of flow stage and solid discharge. Panel (a) shows flow stage as measured in the middle**
**section. Panel (b) shows outlet solid discharge (red bars) as compared with inlet solid discharge (blue horizontal line). It is worth recalling that these measurements refer to a different experiment from that presented in Sect. 3.1.**

    The three sediment pulses that result from major destabilizations in the storage area are all characterized by the same composition (Fig. 7a): a front made of the coarsest fraction of the sediment mixture, a body that exhibits a predominance of sand and a tail characterized by a wide grain size distribution (Fig. 7b). This varying grain size distribution mainly results from
the processes that occur in the storage area. The front made of the coarsest particles constituting the deposit surface ($D_{84} =$ 12.12 mm in average from all front's samples) is inherited from the coarser grains being the first ones to be destabilized in the storage area. These coarser grains always precede the peak of solid discharge, and are materialized in the flow stage measurements by a small bump preceding the main pulse's peaks (Fig. 6a). On the opposite, the sand, which is initially hidden below the surface in the storage area, only emerge and is transported towards the channel when the bulk mass is destabilized.
This large destabilization constitutes the flow stage peak, which exhibits finer grains ($D_{84} = 7.43$ mm) and the highest concentration of sand (33 % by weight). The falling limb of the sediment pulse is composed of a wider grain size distribution ($D_{84} = 7.85$ mm) with a high percentage of sand as well (40 % by weight), but with a decreased solid discharge as a result of the next aggradation phase starting to store sediments in the storage area. This peculiar composition is absent in the second solid discharge peak, where all the samples exhibit an average $D_{84} = 8.63$ mm with little inter-samples variations.

The video recorded one meter upstream of the middle section (see Video 6 in Supplement) allows us to characterize the transport mechanics associated with each part of the pulse. The pulse's front exhibits typical bedload dynamics with grains saltating, rolling, and sliding on the bed (see the first 15 s in Video 6). The coarsest fraction occasionally gets stuck and forms small lateral clusters, consistent with transport for these large grain sizes occurring near the threshold of motion (see $Sect.$ 2.2). These bedforms are ephemeral since sudden impacts of grains can destroy their structure incorporating them in the main flow,

causing the motion of the biggest elements constituting the front to be quite intermittent. The pulse's body is conversely characterized by an enhanced mobility. Our instrumental equipment does not allow us to deeply investigate the nature of the interactions occurring in this dense granular flow (i.e. collisional or frictional, sensu (GDR MiDi, 2004)), but an important role in the transition between the dynamics of the front and that of the body seems to be played by the sand input, since the change in mobility arises when fine particles enter the channel (around $t = 0{:}0{:}22$ in Video 6). Although the grain size distribution

is mainly imposed by the storage area, the pulse's body is also subject to in-channel grain sorting: fine sediments percolate to the subsurface while bigger grains are pushed upward and roll over them. Despite having the same size, we observe that the velocity of these elements is almost doubled compared to the particles constituting the front, and we advance that size segregation is the driving mechanism for this enhanced mobility. It is worth noting that as a result of this process, a portion of the coarse upper layer of the body can eventually move ahead and reach the already developed front before it reaches the outlet

section. That is why the first samples exceeding a value of 200 g s$^{-1}$ of each sediment pulse, despite being considered part of the pulse's body because of the high solid discharge, are characterized by a consistent portion of coarse grains. As the solid concentration decreases, the tail of the sediment pulse is no more congested and is characterized by a saltation dynamics ($t = 0{:}0{:}35$ in Video 6). As opposed to the front, which has comparable solid discharge values, the tail of the pulse is also composed of fine grains. As a consequence, thanks to an enhanced transport capacity (Wilcock et al., 2001; Curran and

Wilcock, 2005), the coarsest fraction of the mixture moves relatively fast. This varying dynamics is missing for the second solid discharge peak, which exhibits a constant bedload dynamics (see Video 7 in Supplement).

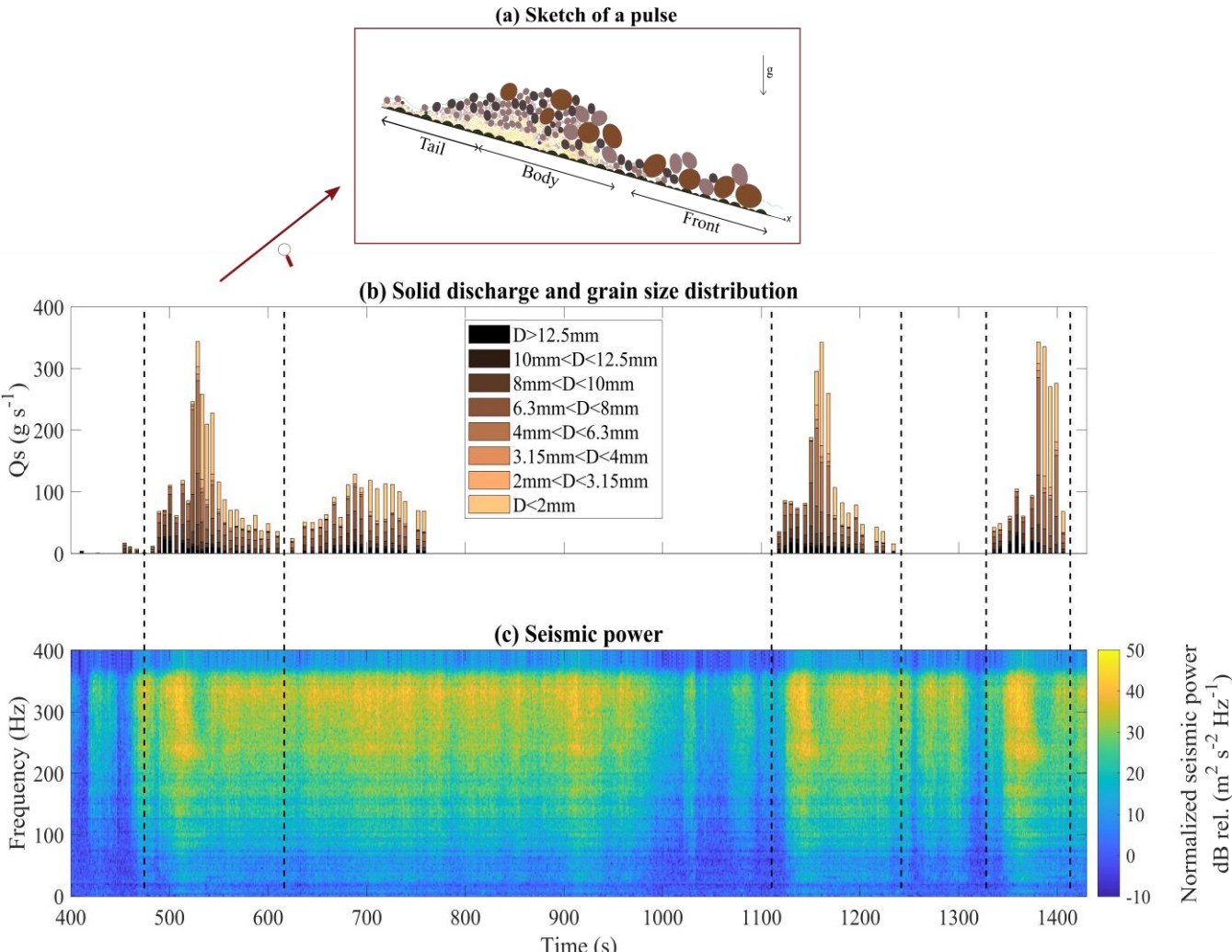

Figure 7: (a) Sketch of the sediment pulse. It can be divided in three parts: a front, a body and a tail. (b) The four sampled pulses and the small solid discharge peak are presented with their grain size distribution. Each coloured bar refers to the particle diameter displayed in the legend, while the bar length is proportional to the percentage in weight of the related particle size. (c) Seismic power detected in the middle section of the flume. The seismic power is normalized with the mean seismic power computed under no sediment transport conditions, and it is shown as a function of time and frequency, where different colours refer to different level of power.

## 3.3 Pulse-induced seismic motion

The passage of sediment pulses is associated with significant increases in seismic power over the whole frequency range, with the highest variations occurring above 200 Hz and being of about 30 dB (Fig. 7c, e.g. $t = 500$ s to $t = 1000$ s and $t = 1100$ s to $t = 1450$ s). Comparing the outlet solid discharge samples and the spectrogram (Fig. 7) we observe that seismic power varies considerably during the sediment pulse. Highest mean power always corresponds to the passage of the front, while the

body and the tail are comparatively associated with much lower values (respectively −9 dB and −6 dB compared to the front). We verify that highest seismic power is indeed exclusively due to the passage of the pulse's front thanks to video recordings, on which we observe that (i) most of the channel is occupied by the front and the sediment pulse body is not yet present when the peak of seismic power is reached, and (ii) seismic power starts decreasing when front's particles get out of the channel. Similarly, the seismic signature of the second peak solid discharge is characterized by a high level of seismic power above 200 Hz, but as opposed to that of bigger sediment pulses, seismic power is proportional to solid discharge, with higher seismic power in the $200 - 300$ Hz frequency range during the passage of higher solid discharge.

## 4 Discussion

### 4.1 The impact of the experimental conditions on the behaviour of the upstream storage area

Here we discuss the extent to which the geometric specificities of the storage area (e.g. the size and the slope of the basin) as well as the boundary conditions (i.e. the input discharges) may have an impact on our observations. The size of the storage area controls the maximum volume of the deposit. A bigger size takes longer to fully fill before the deposit approaches the downstream channel and destabilize, therefore longer periods of aggradation are expected. In such a case, the magnitude of the erosion phase (i.e. the eroded volume) might be bigger given the larger surface exposed to the flow. By contrast, a smaller storage area might mean more frequent but smaller destabilizations. Similar implications are expected through varying the basal slope of the storage area, since higher slope would exert stronger stresses on particles due to gravity, likely leading to more frequent and smaller destabilizations, and vice versa. Different inputs of liquid and solid discharges may also have an impact on the frequency and magnitude of destabilization cycles, the former by changing the stress on the surface particles, the latter by affecting the rate of aggradation. Thus, we believe that the frequency and magnitude of aggradation and erosion phases are mainly set by the geometry of the storage area and the boundary conditions. As a result, we avoid interpreting these aspects and concentrate our analysis on the processes associated with destabilization.

### 4.2 The control of the finest fraction on the *en masse* destabilization of sediment accumulation zones

This experimental setup has been designed to investigate if a self-formed deposit could generate sediment pulses for a downstream channel. We find that the bimodal deposit (Main experiment) exhibits a pulsating behaviour, i.e. self-induced alternating phases of storage and release of sediments under steady external forcing. In our experiments, the period of each cycle is likely dependent on deposit's surface slope variations, since the major destabilizations of the deposit always occur within a small range of longitudinal gradient (48 % $\pm$ 3 %) and the following aggradation phases as well (23 % $\pm$ 1 %).

However, we suggest that the dynamics of these alternating phases is mainly controlled by the presence of a fine fraction (sand in our experiments) and its downward percolation.

While kinematic sieving stabilizes the deposit during the aggradation phase through building a coarse armour on the surface as observed in alluvial beds (Recking et al., 2009; Bacchi et al., 2014), the presence of sand in the subsurface not only triggers but also enhances *en masse* erosion. We link the triggering mechanism to a decrease in the deposit's hydraulic conductivity: when sand moves downward in the mixture, it fills the interstices between grains and obstruct the subsurface water flow; as water can hardly infiltrate, a surface flow develops and starts increasing shear stresses on the particles constituting the armour, which is consequently prone to instability when a certain slope is reached. The effect of fines on the hydraulic conductivity of a sediment deposit and its failure has been investigated by Hu et al. (2017, 2018) with flume experiments on the initiation of flow-like landslides. The authors show that the low hydraulic conductivity of mixtures rich in fines (called in the cited papers as "small particles" to underline their non-cohesive nature) promotes pore pressure's build-up and the consequent failure of the granular deposit. Similarly, fines' availability has been proposed as a factor able to lower the threshold of debris flow initiation from loose sediment deposit for increasing pore water pressure (Baer et al., 2017). Since our experimental equipment does not allow to estimate pore pressure, we cannot draw conclusions about its potential increase upon failure. However, the video recording makes us hypothesize that surface water flow exerts a major control on the destabilization process. We do not observe a well-defined slope rupture of the soil but rather the disintegration of deposit's armour that slides downstream under drag forces (e.g. $t = 0{:}05{:}32$ or $t = 0{:}08{:}25$ in Video 1 in Supplement). It is only at a later stage that the incision deepens due to the formation of a channelized flow (e.g. $t = 0{:}06{:}45$).

Thus, we propose that large parts of deposit's armour fail *en masse* once the deposit is destabilized thanks to the percolated sand, that acts as a carpet over which the overlying grains slide. This "granular lubrication" effect has been reported in previous works, where small particles are shown to increase the run-out length of granular avalanches (Linares-Guerrero et al., 2007; Phillips et al., 2006) and the mobility of granular column (Lai et al., 2017). Interestingly, Hu et al. (2017) wonder if the viscous interface between water and small particles could affect the flow-sliding: our observations on granular lubrication can be seen as additional evidence supporting their intuition. Changes in pore pressures occurring after soil's failure have also been shown to help debris flow mobilization through decreasing its frictional strength until liquefaction (Iverson, 1997; Iverson et al., 1997). Although this process could help destabilization in our experiments, we believe that its effect is not major since the armoured surface is made of coarse grains ensuring relatively efficient drainage conditions, and thus likely preventing large pore pressure build-ups. Iverson et al. (1997) point out that the transition from localized failure to wider and generalized sediment flow might also occur without contraction (i.e. without additional pore pressure variations) if the mass becomes agitated enough through developing granular temperature while moving downslope, which may also occur in our case.

The experiments using the uniform coarse material and the bimodal mixture characterized by a low fraction of sand (Run R2 and Run R3, respectively) support our hypotheses since for equal boundary conditions the deposit shows a much inhibited mobility without any releases to the channel. Run R2 is characterized by a high hydraulic conductivity, and the deposit behaves like a dry granular pile with small grain avalanches that barely spread over the storage area. Run R3 is characterized by the

development of a limited surface water flow, and a single destabilization with an extremely confined run-out ($t = 0: 03: 45$ in
       Video 4) with no channelized flows eroding the mass.

       Although the processes that drive the massive failure of sediment accumulation zones may be many, the presence of a fine
       fraction seems to be the common denominator. Therefore, we propose that the granulometric composition of deposits should
       be carefully taken into account to assess their propensity to abruptly evacuate material to downstream channels. We
acknowledge that direct field measurements are often difficult to carry out in the upper part of mountain catchments, but
       geological maps and high-resolution topographic surveys (Loye et al., 2016) could be sufficient for a diagnostic analysis on
       grain size distribution, as the amount of small sized fraction mostly depends on the local lithology and type of mass wasting
       processes involved in sediment production (e.g. fragmentation in rock avalanches (Zhang and McSaveney, 2017) and
       landslides (Davies and McSaveney, 2009)).

### 4.3 The dynamics of sediment pulse's body as set by the sand input from the storage area

       Our experiments show that the sediment pulses travel downstream with ephemeral interaction with the bed, since the channel
       is completely free of sediments after the passage of the pulse's tail. Here we would like to stress how the massive input of fine
       particles during the upstream erosion phase influences the dynamics of the pulse. While at the beginning the sediment pulse's
front is characterized by an intermittent dynamics and a reduced velocity, the motion of the biggest particles is dramatically
       enhanced with the body's arrival and passage. Over one century ago Gilbert (1914) demonstrated that the introduction of fine
       particles could enhance the transport efficiency of a mixture, and many works investigated this process experimentally
       (Wilcock et al., 2001; Curran and Wilcock, 2005), but only recent experimental studies underline the role played by grain
       sorting (Recking et al., 2009; Bacchi et al., 2014; Dudill et al., 2018; Chassagne et al., 2020). Whereas Bacchi et al. (2014)
and Dudill et al. (2018) show that fines enhance the mobility of big particles by smoothing the surface where they move,
       Chassagne et al. (2020) propose from numerical modelling that after percolation fines can create a "conveyor belt" transporting
       at higher velocity the overlying coarse grains. Although the authors showed that an exclusive "conveyor belt" contribution on
       the increased mobility of larger grains implies a net separation between the two main sizes, which is missing in our experiments
       since particles are quite mixed on the surface, from the video recording big particles appear to be passively transported
downstream over a fast layer of small grains (blue pebbles over a yellowish carpet from $t = 0: 0: 25$ to $t = 0: 0: 32$ in Video
       6 in Supplement). These observations lead us to suggest that the efficiency with which the pulse's body is digested by the
       channel without leaving any trace mainly depends on the capability of fine particles to carry coarser particles as a result of
       grain sorting, rather than hydrodynamics.

### 4.4 Similarities with debris flow events

Although the experimental conditions investigated in this study do not specifically concern debris flows according to classical
       criteria that rely on sediment concentration (the maximum sediment concentration of our pulses does not exceed the commonly

adopted threshold of 50 % by volume) and driving forces involved (the channel's slope does not allow a gravity-driven mass movement), sediment pulses' dynamics exhibits remarkably similar characteristics to those of stony debris flows (Takahashi, 2014). A first similarity consists in the granulometric composition: a front made of boulders, a body characterized by a wide grain size distribution and a much more diluted tail (Iverson, 1997; Stock and Dietrich, 2006; Takahashi, 2014). To our knowledge this feature has been exclusively associated with processes occurring in the transportation zone such as in-channel size segregation (Iverson, 1997). Although we observe this latter process as well, our experimental work shows that a selective entrainment of grains also occurs in initiation zones, which can then have a significant role for influencing the textural composition of downstream propagating pulses. Given the difficulty of carrying out direct field observations in initiation zones (Berti et al., 1999; Imaizumi et al., 2006; McCoy et al., 2012; Loye et al., 2016), we suggest that this kind of experimental setup could be useful for investigating the mechanisms of both debris flow initiation and transportation.

Our findings also confirm the hypothesis of Kean (2013) for which the presence of a sediment accumulation zone can play a key role in the triggering of cyclic debris flow surges resulting from alternate aggradation and mass failure phases. In particular, the authors point out that the regressive instabilities (sensu Zanuttigh and Lamberti, 2007) of those debris flows that are generated by water runoff (i.e. runoff-induced debris flows) may develop thanks to the presence of local low-slope sections of the channel where sediments can temporally be stored and then suddenly released. Channel portions characterized by a local decrease in sediment transport capacity, referred to as "sediment capacitors", can turn steady or quasi-steady supply conditions into discrete debris flow pulses. In modelling this phenomenon, Kean et al. (2013) use a uniform grain size distribution but acknowledge that a wide grain size distribution might affect surge characteristics. Our experiments corroborate this consideration and further stress how the granulometric composition of deposits can exacerbate the debris flows' pulsating behaviour.

## 4.5 Links between pulse's dynamics and seismic noise

We observe a complex seismic response to sediment pulses, characterized by a non-unique dependency of seismic power on sediment transport characteristics such as grain size and sediment flux. Highest seismic power is caused by the propagating front, consistent with the presence of larger grains causing more energetic impacts (Tsai et al., 2012). However, reduced seismic power is observed during the passage of the pulse body, although this latter is associated with the highest sediment flux, a parameter which is often aimed at being inverted from the seismic signal (Tsai et al., 2012; Bakker et al., 2020). Using the prediction of Tsai et al. (2012) that seismic power approximately scales as $D_{94}^3 q_s$, where $D$ is the particle diameter and $q_s$ is sediment flux, we find that the reduced seismic power of 9 dB between the front and the body of the pulse cannot be explained solely by changes in $D$ and $q_s$, since $D$ decreasing by about a factor of 0.7 ($D_{94} = 12.93$ mm for the front compared to $D_{94} = 9.32\ mm$ for the body) and $q_s$ increasing by about a factor of 4 (from 80 g s$^{-1}$ for the front up to 340 g s$^{-1}$ for the body) would yield approximately constant seismic power. Since seismic records show a reduced sensitivity to the pulse's body, which in fact accounts for the largest fraction of the sediment flux, the capability of existing models of reliably inverting solid discharge from seismic power is questioned for this kind of transport processes.

Since our sediment pulses show similarities with debris flows (see *Sect.* 4.4), we find appropriate to compare our observations also with expectations from theories of debris flow-induced seismic noise. Conveniently, the limited channel length in our experimental setup allows us to study the seismic responses of the three different parts of the pulse (front, body, and tail) separately, since when one component of the pulse acts the other one is not yet on the channel or has already left it. On the contrary, in the field all parts of the pulse can potentially contribute to the overall measured seismic noise, such that the drop

in seismic power observed in our experiments during the passage of the body could be "hidden" in the field by the seismic noise induced by a louder upstream tail and downstream front. Our observations are consistent with most field surveys and models, for which the front (sometimes referred to as snout) generates a stronger seismic power than the following flow as it carries the largest clasts (Arattano and Moia, 1999; Lai et al., 2018; Coviello et al., 2019; Farin et al., 2019; Allstadt et al., 2020). However, the relationship between seismic noise and flow thickness is contrasting. While some observations show a

good correlation between flow thickness and fluctuating basal stresses (Allstadt et al., 2020) and some models reveal no or rare direct dependence (Lai et al., 2018; Farin et al., 2019), our experiments show a clear negative correlation since pulse's body is characterized by the peak of flow stage (Fig. 6). According to Cole et al. (2009) and Allstadt et al. (2020), this could be explained by body's high solid concentration. Indeed, they observe a negative correlation between bulk density and seismic noise, and therefore propose that more agitated flows are "louder" than denser and plug-like flows. This interpretation would

be also consistent with the increase of seismic noise associated to the pulse's tail, which is again much more diluted than the body.

Further work remains to be conducted in order to fully unravel the control of the pulse's internal dynamics on the generated seismic noise. In particular, it appears as essential to more quantitatively investigate the effect of grain sorting, which likely plays a crucial role through pushing upward the biggest particles, thus preventing them from directly impacting the bed and

485 reducing their contribution to seismic noise. This would be consistent with the field observations of Kean et al. (2015), who suggest that the presence of a sediment layer over the bedrock can strongly damp the seismic signal generated by a debris flow. Detailed analysis of particle impact velocities, rates and applied forces across the different grain sizes and the different pulses components would help further addressing these aspects.

**5 Conclusions**

We carry out flume experiments characterized by an original setup where instead of feeding the flume section directly as usually done, we supply with liquid and solid discharge a low slope storage zone acting like a natural sediment accumulation zone and connected to a 18 % steep channel. The experiments reveal that:

1) under constant feeding conditions, when a bimodal grain size distribution with a high fraction of sand is used, the
495 storage area is subject to alternating aggradation and erosion phases. The high morphological mobility of the deposit is due to several autogenic processes, but the presence of sand appears to play a key role. In particular, if during the

aggradation phase grain sorting enhances the stability of the deposit in coarsening its surface thanks to the downward percolation of the fine particles, we propose that the infilling of the subsurface with fine material contributes to the destabilization of the deposit by two means: (i) it reduces the hydraulic conductivity of the deposit and causes the formation of a significant surface water flow that in turn increases the stresses over the armoured layer, (ii) it acts like a smooth carpet where the coarser grains slides *en masse*.

2) the erosion phases correspond to the generation of sediment pulses towards the downstream channel. The evolution of the sediment deposit affects not only the magnitude of the sediment pulses, but also their rheology and dynamics. When major destabilizations of the sediment deposit occur, each sediment pulse can be divided in three different components as follows: a front having a low solid discharge made of the coarsest fraction of the sediment mixture, inherited by the destabilization of deposit's surface; a body that corresponds to the peak of solid discharge, composed of a high concentration of sand coming from deposit's subsurface; a tail characterized by a low solid discharge and a wide grain size distribution, with sediments still transported while the next aggradation phase starts to develop in the storage area.

3) pulses in sediment transport can be detected by seismic measurements. We find that the sediment pulse's front dominates the overall seismic noise. However, we report a complex link between seismic power and the different parts of the sediment pulse, which questions the validity of current models and theories to such transport dynamics. Further work is needed to unravel the role of the different pulse's geometrical and dynamical parameters on the generated seismic noise.

From a practical point of view, these results have strong implications in natural risk management. First, we show that the proximity of upstream sediment accumulation zones must be considered a potential source of sediment pulses for mountain rivers, regardless of bed sediments' availability. Second, since the grain size distribution is shown to have a direct influence on the mobility (i.e. stability) of debris deposits, we challenge the classical approach for which the *sediment budget* of mountain catchments is merely reduced to an available volume and hydrological conditions are considered the main factor controlling the activation of external sediment supply. Instead, the granular conditions of deposits that are coupled with mountain streams or stored in low slope portion of the channel should be taken into account for assessing the occurrence and dynamics of such dramatic transport events. Finally, our seismic findings challenge the application of current theoretical frameworks to invert bedload flux from the seismic noise associated with this kind of transport processes.

*Author contributions*: MP, AR and FG designed the laboratory experiments. MP developed the flume and HB led the installation of the instrumental equipment. MP carried out the experiments with the help of HB. MP interpreted the results, with input from FG, AR and HB. MP led the writing of the paper, and FG and AR revised and contributed to it.

*Competing interests*: The authors declare that they have no conflict of interest.

*Data availability*: The datasets analysed during the current study are available and temporary shared on this link: https://www.dropbox.com/sh/n3oqyl06csmnipg/AABnNuhwblRZIAhSRFt6EbEba?dl=0

In case of acceptance we plan to give access to all data through the data repository platform Zenodo with an associated Digital Object Identifier (DOI).


*Video supplement*: The videos used for the analysis are temporary available on this link: https://www.dropbox.com/sh/ctyxz64hn0di41s/AAAgvoEkXihVE44norGPC-E8a?dl=0
The DOI's generation through the AV Portal of TIB Hannover is in progress.

*Acknowledgements*: This research is funded by the French Agence nationale de la recherce, project ANR-17-CE01-0008. We acknowledge the support of the INRAE Research Centre of Grenoble for the laboratory and the instrumental equipment. We thank Christian Eymond-Gris, Frédéric Ousset, and Xavier Ravanat for the technical support in the development of the flume. We thank Maarten Bakker for helping in the analysis of the seismic data. We thank Guillaume Piton for the constructive discussion on the experimental scaling of the flume.

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
