# Peer review of "S1 Introduction"

_Earth Surface Dynamics, 2021_

## Author Comment (AC1)

Response to Reviewers' Comments for:

"Genesis and propagation of exogenous sediment pulses in mountain channels: insights from flume experiments with seismic monitoring"

**Comments from Referee #1**

This study presents the results of a unique set of experiments to understand the formation mechanisms of sediment pulses in steep streams. The manuscript is very well written, and the quality of the measurements appear to be excellent. The study draws interesting connections with debris flows and reaches important conclusions on the interpretation of ground vibrations that will make the study of interest to a broad audience.

We thank the reviewer for his thorough review, providing constructive and detailed comments to help improving the quality of the manuscript.

I think one aspect of the study that could use more discussion is the periodicity of the sediment pulses. What sets the frequency of the pulses?

The reviewer legitimately suggests more discussion on (1) the periodicity of sediment pulses and (2) the process that triggers the reduction in transport capacity and a new phase of aggradation (comment made below by the reviewer, addressed to line 177 of our manuscript), two aspects that are in close connection. In our experiments, the period of each cycle is likely dependent on deposit's surface slope variations, since the major destabilizations of the deposit always occur within a small range of longitudinal gradient (48 % $\pm$ 3 %) and the following aggradation phases as well (23 % $\pm$ 1 %). The slope seems to play a role in both phases by governing the transport capacity of the flow. We now further elaborate on these aspects in the main text by giving more information in the Results, Section 3.1 (lines 223-235), and in the Discussion, Section 4.2 (lines 363-365). However, despite the process being periodic by nature, we decided not to focus specifically on the frequency and magnitude of pulses because we believe that they are mainly set by the specifics of our experimental setup (geometry and specifics of the storage area) and the boundary conditions (inlet liquid and solid discharge). Indeed, destabilizations result in sediment pulses only when the deposit is well connected to the downstream channel. In response to the reviewer comment we now have added a specific section in the discussion (section 4.1) in which we explain in which ways our experimental conditions may affect the frequency and amplitude of pulses.

41: Consider also citing earlier work on this topic, like Wiberg & Smith, WRR, 1991

We thank the reviewer for the suggestion. We have added the proposition of the reviewer and an earlier paper on the topic (see lines 38-39):
Bathurst, J. C., W. H. Graf, and H. H. Cao (1982), Initiation of sediment transport in steep channels with coarse bed material, paper presented at Euromech 156: Mechanics of Sediment Transport, Eur. Mech. Soc., Istanbul, Turkey, 12–14 July.

80: "however the extent to which they continue to apply to flows like sediment pulses remains to be investigated" True, but there are a number of studies, some of which you mention, that have applied those methods debris flows, which are an extreme form of a sediment pulse.

The reviewer is correct that mechanistic models have been built to study debris flows as well. In particular, very recent works have tested that of Farin et al. (2019) in the laboratory (Arran et al., 2021) and in the field (Zhang et al., 2021). We have added the two later citations that, due to their recent publication, were missing in the original manuscript (lines 82-83). As suggested by the reviewer, we now emphasise more clearly that sediment pulses may lie in between the two extremes of the spectrum of sediment transport (lines 83-85)

96: What is the slope of the upstream storage area, 0 %?

The slope of the upstream storage area is ~0-1%. We have added this information in the manuscript in Section 2.2 (see lines 101-102). We also now incorporate a discussion on the potential effect of slope on the observed processes (see section 4.1, lines 349-352).

118: What is the sampling rate of the ultrasonic stage sensor?

The sampling frequency of the ultrasonic stage sensor is 100 Hz. We apologize for not having stated this in the previous version of our manuscript. We have added this information lines 119-120 of the revised manuscript.

177: Please elaborate on the process that triggers the reduction in transport capacity and new phase of aggradation.

We particularly thank the reviewer for the question. It has led us to improve the investigation on the processes occurring in the storage area. As stated above while responding to the general comment of the reviewer about the periodicity of pulses, the deposit reaches an average minimum slope of ≈ 25 % at the end of the erosion phase. At this moment, the transport capacity is at its minimum. Although some releases of sediments are still possible, a new armouring phase starts developing with the formation of bars made of coarse particles, which make a new aggradation possible as the flow becomes shallow and unchannelized. We have better developed the subject through adding dedicated text in Section 3.1. (lines 226-231)

179: How many aggradation/erosion phases do you typically see in a run? And how long is a typical run? Or, what is the period of each cycle?

We apologize for not having reported the experiment's duration, which is typically about 30 min. We have added this information to the manuscript (Table 1). We typically observe four aggradation/erosion phases per 30 minutes of experiment. We have added this value as well (lines 233). We also now provide a discussion on what we think mainly controls these aspects (Section 4.1).

179: "Erosion's intensity is characterized…" I do not follow this sentence.

With these lines we attempted to describe the shape of the peaks in figure 5. Following the reviewer's comment, we realize that the sentence does not provide additional information compared to what already described. As a consequence, we have removed these lines.

181: "presence of sand and their percolation" Please clarify if you are referring to the percolation of the fine sand through the coarse material, or the percolation of water through the deposit. I think you mean the former, but the statement could be misinterpreted.

Yes, with this sentence we refer to the percolation of fine sand. We reformulate the sentence to avoid misunderstandings (lines 267-268)

226: "Given the different genesis…." I do not follow this statement about why the second solid discharge is not considered a sediment pulse.

We thank the reviewer for this comment that allows us to reconsider our original choice not to treat the second solid discharge peak as a sediment pulse. As stated in the manuscript, sediment pulses' dynamics is controlled by the processes occurring in the storage area. After large erosion phases and before a new aggradation phase, there exists a time window during which the volume doesn't vary significantly (indicated by black arrows in Figure 5a). The occurrence of each new aggradation phase is indeed not trivial since right after generating sediment pulses, the deposit is well connected to channel's inlet section. During this intermediate phase, a non-negligible amount of sediments can still reach the downstream channel through small but significant channelized flows in the deposit: this is what generates the "second solid discharge peak". Its genesis is thus quite different from the other three sampled sediment pulses because it is not the result of a major erosion phase with armour breaking, but rather represents a nearly constant release of material from the storage area. However, since in the Introduction sediment pulses are defined as "disturbances in bed elevation that propagate downstream translating as a coherent wave end/or dispersing in place", this second solid discharge peak falls into this category. We precise this nuance in the manuscript accordingly by the added text lines 280-282.

261: "the tail of the sediment pulse is back to a saltation dymanics" Please clarify. The sediment transport processes in the tail transition from what? to saltation.

The tail of the sediment pulse transitions from congested flow of the pulse's body back to a saltation dynamics. We have reformulated the sentence to clarify this point (line 317)

296: "as water cannot infiltrate" Water can infiltrate sand. The addition of fine grain particles reduces the hydraulic conductivity.

The reviewer is right. We have implemented the suggested clarification (line 372)

299: referred to in the papers… Which papers? This paper? The sediment transport literature?

We refer to the papers cited just before. We reformulate the sentence to avoid misunderstandings (line 375)

307: Could elevated pore pressures from contraction during motion also aid in "granular lubrication" like it does for debris flows?

We thank the reviewer for this interesting comment. Since our experimental equipment does not allow us to monitor pore pressures variations inside the deposit, pore pressures-induced phenomena are only partially discussed in the manuscript (Section 4.2). The present comment from the reviewer has encouraged us to extend the literature on the subject, pushing the analogy with debris flows as suggested. Changes in pore pressures during mass failure have been proposed to aid debris flows mobilizations, especially for landslide-triggered debris flows (Iverson, 1997; Iverson et al., 1997). Indeed, the transient increase of pore pressures during contraction may weaken the mass by reducing the frictional strength of the soil. However, we believe that the *en masse* failures occurring in our experiments are rather mainly driven by the subsurface sand slope that makes the armour extremely unstable while subject to water flow. This interpretation is supported by analysis of -the video recordings (Video 1 in Supplement), on which we observe that the initial destabilization involve a layer of $2 - 3D_{50}$ constituting the armour, while only at a later stage the incision deepens due to the formation of a channelized flow. The coarse grain size distribution of the surface makes us hypothesize drained conditions without significant pore pressure changes during motion. Moreover, in contrast to Iverson et al. (1997), we do not observe a well-defined "slope rupture" of the soil but rather the disintegration of deposit's armour that slides downstream. Interestingly, Iverson et al. (1997) point out that the transition from localized failure to wider and generalized sediment flow might also occur without contraction (i.e. without additional pore pressure variations) if the mass becomes agitated enough through developing granular temperature while moving downslope, which may also be our case. We have incorporated these thoughts in the discussion section of our manuscript, lines 389-395.

315: I suggest omitting conclusions based on experiments that are not presented in this paper. If they are important observations for this study, shouldn't the experiments be included?

We follow the suggestion of the reviewer. We further dedicate a specific section of the manuscript on the influence of the experimental setup on the observations (Section 4.1)

385: Please elaborate how your setup allows you to differentiate the source of the signal in a way that cannot be accomplished in the field. Would a seismic sensor very close to a natural channel have similar measurement capabilities as your laboratory setup?

The lines commented by the reviewer recall also Section 3.3 (lines 336-338), where we explain how we verify that highest seismic power is exclusively due to the passage of the pulse's front. We believe that the main difference between our experimental setup and the field is the presence of an outlet section. Indeed, thanks to the video recording we verify that the seismic power starts decreasing when front's particles get out of the channel, and this helps us discriminate the contribution of the front from that of the body.

Nevertheless, the reviewer is correct that the distance between a potential seismic sensor and the natural channel is definitely an element to consider to evaluate its ability to differentiate signal's sources. In the field, a geophone is expected to remain sensitive to the front even during the passage of the body especially if not installed in the immediate proximity to the river. In such a case, the drop in seismic power observed in our experiments could be "hidden" by the seismic noise induced by a front further downstream. The low value of seismic power associated to the body is even more emphasized by the following tail, which

generates more vibrations despite having a much lower solid discharge. Again, and for the same reasons as described above, we suspect that this discrimination might be difficult to accomplish, because all parts of the pulse can potentially contribute to the overall measured seismic noise. We have added some lines in Section 4.5 to moderate our initial statement and clarify the potential differences capability of the seismic monitoring between field and laboratory (lines 369-371).

**Comments from Referee #2**

The paper presented by M. Piantini and colleagues deals with the sedimentary pulses that occur in mountain streams. The authors deal with the phenomenology of these pulses as well as their possible mode of study by acoustic and seismic methods. The main technique used by the authors is the simulation of the phenomenon in an artificial channel equipped with an original sediment supply system and sound sensors installed throughout the canal.

We thank the reviewer for taking the time to review the manuscript and providing a criticism which helped improve our manuscript.
We must clarify a potential misunderstanding here. As opposed to what is stated by the reviewer, in this paper we do not consider acoustic methods and sound sensors, although those have been deployed for monitoring bedload (Geay et al., 2017). Instead, we deal with seismic techniques, which consist in analyzing seismic waves propagating through channel bed and sidewalls, as done in other experimental works cited in the paper (Gimbert et al., 2019; Allstadt et al., 2020). Three sections of the channel are equipped with a remote transducer ultrasonic sensor and a geophone to respectively measure flow stage variations and detect flow-induced flume motion.

This paper lacks an important discussion which concerns the transposition of experimental results to nature. It seems necessary to demonstrate that the processes which are observed in the experiment are conceivable in nature and vice versa. Downscaling, for example, is not discussed anywhere in the paper. Likewise, what is the significance of the upstream sediment supply system of the experimental apparatus? The authors describe natural systems in which cliffs bring sediment to streams (Line page). The protocol used seems to be quite different. On the other hand, there is no discussion of the time and space scales of the phenomenon in nature. For example, does the particle size distribution used in the simulations have any significance for the natural examples? In order to be published this paper should clearly address these issues.

We thank the reviewer for the comment, we appreciate the core of this criticism. Before addressing specifically each concern raised by the reviewer, we would like to emphasize that with our experiments we do not aim to reproduce a particular natural prototype. Instead, we aim to focus on processes at the bulk scale in an "analytical framework" (Paola et al., 2009). Our experimental setup is thus meant to be "a scaled version of a general geomorphic feature" (Peakall et al., 1996), whose natural context have been described through a consistent part of the Introduction (lines 47-60).

That said, however, we agree with the reviewer that our initial manuscript (i) was misleading on the relevant processes we meant to represent in the storage area, and (ii) lacked important downscaling aspects regarding flow hydraulics, grain size distribution and spatial and

temporal scales of sediment pulses. In the revised version of our manuscript we have addressed these two weaknesses as exposed below. We believe that with these modifications we are now much clearer in the main text in which ways our experimental setup and our investigated flow and sediment transport conditions satisfy scaling considerations with nature.

I.      Processes in the storage area

Our study refers to steep and small low-order channels that are coupled to sediment production zones (lines 47-50). The activity of headwater sediment sources has been shown to exert a control on the dynamics of downstream rivers (Piton and Recking, 2017). As observed by the reviewer, Figure 1 in the manuscript represents a mountain catchment where sediments are produced by cliffs. However, as stated in the manuscript (line 52), the catchment of the Roize River is simply used as an example, the purpose of Figure 1 being to show how large amounts of sediments can accumulate at slope's toe in the proximity of mountain rivers. Instead of directly reaching closer streams, sediments produced by mass wasting processes are often temporary stored in the form of talus slope or along low-slope stretches as loose scree deposits (Berti et al., 1999; Fontana and Marchi, 2003; Gregoretti and Fontana, 2008). In this sense, the storage area of our experimental setup does not refer to a specific natural setting but aims at representing a zone where sediments can accumulate as a result of material falling from cliffs, rocks rolling down steeper slopes or other processes that we do not study in this manuscript. We prescribe solid and liquid discharge as input in order to let the deposit develop by itself and reproduce the flows that typically feed and destabilize these sediment deposits (i.e. runoff descending from upstream rocky slopes, as the waterfall observed by Berti et al. (1999) below the massive cliffs of the Dolomites Mountains, Northern Italy). We consider the choice of having a self-formed deposit particularly original and realistic as representative of the processes occurring in nature: the consistence of the observed processes (e.g. grain sorting and *en masse* destabilizations in the deposit) is discussed in Section 4.2. We believe that the storage area and the whole experimental configuration properly depicts this topographical setting.
We have changed formulation in the Introduction section to better convey the above-listed arguments and avoid potential confusion (lines 47-60). Moreover, following the comment of the reviewer, we have also added a discussion about the potential influence of the experimental setup on the observed processes through a new section (Section 4.1).

II.     Scaling considerations

We have conducted a dimensional analysis in order to verify the hydraulic and sediment transport similarity with natural streams (Peakall et al., 1996; Kleinhans et al., 2014), while the grain size distribution has been chosen with reference to existing experimental works and natural mountain channels (see Section 2.2 in the new manuscript).

1-  Hydraulic and sediment transport similarity

We now provide the dimensionless numbers used to design the experiments in Table 1 of Section 2.2, which are the Froude number, the Reynolds number, and the Reynolds particle number, together with some comments in the associated section of the manuscript. The values of these dimensionless parameters characterize a supercritical, hydraulically rough, and turbulent flow that is consistent with natural mountain streams ($Fr > 1, Re > 2000, Re^* > 70$). (lines 170-173 and Table 1).
Moreover, we incorporate several scaling considerations that have been made for building the

flume and setting the boundary conditions through restructuring Sect.2.2 (lines 138-179). We estimate the scale reduction of the flume by comparing the characteristic particle diameters of the experimental bed with those of two well-documented steep mountain streams, considered as reference natural channels. This geometric scale is then used to upscale our experimental configuration following the guidelines of Peakall et al. (1996).

2- Grain size distribution

The reviewer asks if the grain size distribution (GSD) used in the experiments is coherent with natural examples. Mountain channels are typically characterized by a wide grain size distribution ranging from fine elements to large boulders provided by an external sediment supply which is often bimodal (John Wolcott, 1988; Casagli et al., 2003; Sklar et al., 2017). We therefore choose a GSD with these characteristics and with reference to that used in previous experimental works on steep slope (Bacchi et al., 2014). The GSD of the reference and supplementary experiments are instead used to explore the effect of grain size heterogeneity on the behaviour of the deposit. We add specification on the GSD in Section 2.2 (lines 148-154). However, it is worth mentioning that the fine tail of grain size distributions used in small-scale experiments is often truncated to prevent exaggerated cohesion effect (Peakall et al., 1996) so that a rigorous comparison with real rivers is rarely possible.

3- Spatial and temporal scaling of pulses

As opposed to processes at the bulk scale, we believe that the geometry of the experimental storage area (e.g. the size and slope of the basin) and the boundary conditions (i.e. the input discharges) can have an impact on the spatio-temporal dynamics of destabilization phases. In particular, the experimental setup seems to play a role on the frequency and magnitude of the aggradation and erosion phases. We explore these hypotheses through a new section in the revised manuscript (Section 4.1). For this reason, we do not attempt to investigate on the frequency and magnitude with reference to natural settings. Moreover, concerning the time and spatial scale of pulses in nature, we know that sediment transport in mountain streams is highly intermittent, but existing field observations do not provide adequate temporal and spatial information. This further supports the use of surrogate monitoring technique such as seismic methods, as discussed below. We added a line in the Introduction (line 74) to clarify the lack of field observations, which further motivates the use of seismic observations in our experiments.

In addition, the authors propose a technical study (itself not dimensioned) which could be applied to study transport phenomena. This approach is interesting but makes the discourse complex. It seems to me preferable that the authors concentrate on a single problem (sedimentary pulse or analysis of sound produced by sediments in a channel) and reserve the other for another article.

We thank the reviewer for the comment. We believe that the clarifications presented above and the modifications added to the manuscript now adequately show that (i) the technical study is well dimensioned and (ii) the processes are consistent with those expected in nature. The reviewer proposes to remove the part that addresses the sediment pulse-induced seismic motion. However, as being part of the measurements protocol, we strongly feel the seismic analysis is an integral part of this study by two means. On the one hand, seismic methods are

shown to provide unique information about sediment pulses dynamics. As stated in the Introduction (lines 75-78), sediment pulses are challenging to track in the field and as depicted in Figure 2 only post-event observations are often possible. On the other hand, our results challenge existing theoretical frameworks to invert bedload flux from seismic noise for this kind of sediment transport processes (see Section 4.5). From this point of view, the manuscript provides the basis for further investigations. We must also recall that Referee #1 well highlighted the relevance of the seismic analysis in the present manuscript, which makes our study of interest to a broader audience. As a result, we have decided to keep the seismic aspects as an integrant part of our manuscript.

Bibliography

[revised manuscript text omitted]